

1       **Delamination in Tibet: Deriving constraints from the density of eclogite**

2       Zhilin Ye [a,b], Dawei Fan [a,*], Bo Li [a,b], Qizhe Tang [c], Jingui Xu [d,*], Dongzhou Zhang [d], Wenge Zhou [a]

3       [a] *Key Laboratory of High-Temperature and High-Pressure Study of the Earth's Interior, Institute of*

4       *Geochemistry, Chinese Academy of Sciences, Guiyang, Guizhou 550081, China*

5       [b] *University of Chinese Academy of Sciences, Beijing 100049, China*

6       [c] *School of Information Engineering, Huzhou University, Huzhou, Zhejiang 313000, China*

7       [d] *Hawaii Institute of Geophysics and Planetology, School of Ocean and Earth Science and Technology,*

8       *University of Hawaii at Manoa, Honolulu, Hawaii 96822, USA*

9       * Corresponding authors.

*E-mail addresses*: fandawei@vip.gyig.ac.cn (D. Fan), xujingui@hawaii.edu (J. Xu)
**Abstract**
Tibet, which is characterized by collisional orogens, has undergone the process of delamination or
convective removal. The lower crust and mantle lithosphere appear to have been removed through
delamination during orogenic development. Numerical and analog experiments demonstrate that
the metamorphic eclogitized oceanic subduction slab or lower crust may promote gravitational
instability due to its increased density. The eclogitized oceanic subduction slab or crustal root is
believed to be denser than the underlying mantle and tends to sink. However, the density of
eclogite under high-pressure and high-temperature conditions and density differences from the
surrounding mantle is not preciously constrained. Here, we offer new insights into the derivation
of eclogite density with a single experiment to constrain delamination in Tibet. Using *in situ*
synchrotron X-ray diffraction combined with diamond anvil cell, experiments focused on minerals



(garnet, omphacite, and epidote) of eclogite are conducted under simultaneous high-pressure and
high-temperature   conditions,   which   avoids   systematic   errors.   Fitting   the
pressure-temperature-volume data with the third-order Birch-Murnaghan equation of state, the
thermal equation of state (EoS) parameters, including the bulk modulus ($K_{T0}$), its pressure
derivative ($K_{T0}'$), the temperature derivative (($\partial K_T/\partial T)_P$), and the thermal expansion coefficient
($\alpha_0$), are derived. The densities of rock-forming minerals and eclogite are modeled along with the
geotherms of two types of delamination. The delamination processes of subduction slab breakoff
and the removal of the eclogitized lower crust in Tibet are discussed. The Tibetan eclogite which
containing 40-60 vol. % garnet and 37-64% degrees of eclogitization can promote the
delamination of slab break-off in Tibet. Our results indicate that eclogite is a major controlling
factor in the initiation of delamination. A high abundance of garnet, a high Fe-content, and a high
degree of eclogitization are more conducive to instigating the delamination.
**Keywords:**
Eclogite, Equation of state, Single-crystal X-ray diffraction, Delamination, Tibet
**1.   Introduction**
The evolution of orogenesis is characterized by lithospheric removal during rapid surface uplift,
mantle upwelling, and postcollisional magmatism, particularly in the Central Andes (e.g. Ehlers
and Poulsen, 2009; Schurr et al., 2006), Himalayas (e.g. Jiménez-Munt et al., 2008; Singh and
Kumar, 2009), and Dabie orogen (e.g. He et al., 2011; Zhang et al., 2010).
It is widely accepted that delamination is the most important mechanism of lithospheric
removal. Delamination is induced and accompanied by two major requisites: (a) the density



difference caused by the negative buoyancy of the delaminated lithosphere; and (b) the presence
of a weak lower crust (lower viscosity) that exists between the strong upper crust and lithospheric
mantle. Usually, two types of delamination are believed to occur in orogen development. The first
is the conventional definition of delamination proposed by Bird (1978, 1979), which was used to
interpret the geodynamic evolution of the Colorado Plateau. In this scenario, mantle lithosphere
peels back from the overlying upper crust and is removed entirely, with the rising hot mantle
filling the lithospheric removal zone (e.g. Göğüş and Ueda, 2018; Krystopowicz and Currie, 2013;
Schott and Schmeling, 1998; Sobolev and Babeyko, 2005). A weak decoupling layer, i.e, the lower
crust, is an essential condition in this delamination model, which may be affected by the
rheological behavior of the hydration, thermal, and chemical characteristics of the lithosphere (e.g.
Burov and Watts, 2006; Morency, 2004; Schott and Schmeling, 1998). In addition to conventional
delamination, an alternative delamination mechanism is convective removal based on the
Rayleigh-Taylor-type instability model (Houseman et al., 1981), namely, viscous "dripping". This
model postulates that there is sufficient perturbation in the lithospheric mantle, which is ascribed
to the strong temperature-dependence of typical mantle rheology, without regard to a specific
weak layer (e.g. Conrad and Molnar, 1999; Gorczyk et al., 2012; Houseman and McKenzie, 1982;
Schott and Schmeling, 1998).
All previous studies attribute the gravitational instability process of lithospheric removal to the
negative thermal buoyancy of the cold lithosphere (Conrad and Molnar, 1999; Houseman and
McKenzie, 1982) or density contrast between asthenosphere and mantle lithosphere
(Elkins-Tanton, 2007; Neil and Houseman, 1999). In any case, the density distribution with
lithosphere pressure and temperature (*P-T*) conditions and chemical composition is of vital



importance to understanding the process of lithospheric removal.

The Tibetan Plateau is the most representative and prominent collisional orogens. Two types of

delamination are proposed to proceed (e.g. Chung et al., 2005; Houseman et al., 1981; Molnar et
al., 1993; Platt and England, 1994; Sun et al., 2020): lithospheric mantle removal and thickened
eclogitized crust removal. The Neo-Tethyan oceanic subduction, India-Asia collision, and Indian
continental subduction could be further considered responsible for the abnormal thinning of the
mantle lithosphere under Tibet (Chung et al., 2005; DeCelles et al., 2011; Li et al., 2019; Ma et al.,
2017; Xu et al., 2008; Zhao et al., 2020). The lithospheric removal event in Tibet corresponds to
Neo-Tethyan oceanic slab break-off. The mechanism is primarily based on density contrasts
between the denser mantle lithosphere and the lighter underlying mantle. Some models reveal that
lithospheric removal is induced by the retreating high-density eclogitized lithosphere detached
from overlying low-density crust (Faccenda et al., 2009; Li et al., 2016; Ueda et al., 2012). Other
alternative models indicate that thickened eclogitized crust is a potential factor deriving
lithospheric removal because the eclogitized crustal root is denser than the underlying mantle and
tends to sink (Krystopowicz and Currie, 2013). Regardless of the above types of delamination, the
density of eclogite is closely related to delamination. Therefore, Tibet provides an excellent
opportunity to understand the role of eclogite density in the process of delamination.

An immense amount of concrete research has focused on the origin and appearance of

lithospheric mantle removal from different angles, such as geophysical (Ren and Shen, 2008;
Tilmann, 2003), geological (Chung et al., 2005; Molnar et al., 1993), petrological (Chung et al.,
2005; Turner et al., 1993), numerical and analog experiments (Bajolet et al., 2012; Göğüş and
Pysklywec, 2008; Morency, 2004; Valera et al., 2011). In particular, numerical and analog





experiments are used as prominent methods to simulate the dynamics of delamination (Göğüş and
Ueda, 2018). Of these studies, the density behavior occurring during the delamination process has
also been investigated intensively following thermodynamic (Duesterhoeft et al., 2012; Semprich
et al., 2010), seismic/tomography (Li and Fang, 2017; Matchette-Downes et al., 2019), and
numerical simulations (Gerya et al., 2004; Li et al., 2016; Sobolev and Babeyko, 2005). However,
few studies have systematically illuminated the issue of delamination from the perspective of
eclogite density. Here, we attempt to offer new insights into the derivation of rock density through
the mineral physics method to constrain delamination in Tibet (Ye et al., 2021). Conducting a
single experiment under high-pressure and high-temperature conditions, we obtain the equation of
state (EoS) of the main minerals of eclogite with fewer systematic errors in the experiment.
Furthermore, the newly derived EoS of the main minerals of eclogite, combined with the
published EoSs of the main minerals of peridotite (Ye et al., 2021), geothermal lines, and collected
eclogite mineral compositions, are further used to elucidate a density evolution model during the
delamination process in Tibet. We argue that the EoSs of minerals could be used in a
straightforward manner as new constraints on the construction of the density model. Using a
simplistic calculation setup, in this study, this density evolution model will shed light on the
possibility of delamination during the orogen process.

**2.   Geological background**
The Tibetan Plateau is composed of four terranes from south to north: the Himalaya, Lhasa,
Qiangtang, and Songpan-Ganzi terranes (Fig. 1). The birth of the Himalayas and Tibetan Plateau
is a consequence of the Indo-Asian collision, which began in the early Cenozoic (Hodges et al.,



2001; Rowley, 1998; Wang et al., 2008). The Neo-Tethyan oceanic slab is proposed to have
detached from the Indian lithosphere, and the onset of the Indo-Asian collision (DeCelles et al.,
2002; Kohn and Parkinson, 2002) particularly occurred in the lower part of the Indian and Lhasa
lithospheres. The tectonic evolutionary history of the Lhasa terrane and Tethys Himalayas is
essential for revealing the origin of the Himalayan-Tibetan orogen. The subducting Neo-Tethyan
slab was thrust into southern Tibet approximately 70-65 Ma (Fig. 1b). With the closure of the Neo
Tethyan Ocean, the India-Asia continent collision caused compressional deformation in southern
Tibet (Ding et al., 2003), and a series of collision breakoff events were delineated spanning from
65 Ma to 42 Ma (Chung et al., 2005, 2009; Lee et al., 2009; Leech et al., 2005; Ma et al., 2014;
Zhu et al., 2011, 2015). During this period, the Indian continental lithosphere might have dragged
down to deeper depths during subduction (Chemenda et al., 2000). Meanwhile, slab rollback
accompanied by the southward migration of asthenospheric convection in Tibet changed the
thermal structure of the mantle wedge. The breakoff of the oceanic Neo-Tethyan slab from the
more buoyant Indian continental lithosphere indicated by the eruption of early Eocene Linzizong
volcanic rocks in the Gangdese arc or the cessation of Gangdese arc magmatism occurred at ~45
Ma (DeCelles et al., 2002), which opened a channel for the upwelling asthenosphere (Chung et al.,
2009; Ma et al., 2014; Zhu et al., 2015). Additionally, geophysical evidence of longitudinal wave
($V_P$) tomography is interpreted for the north-dipping high-speed anomaly, which is ascribed to the
deep Indian mantle lithosphere (Li et al., 2008; Liang et al., 2016; Tilmann, 2003). Subsequently,
the subduction of the Indian continental margin continues at a low subduction angle beneath the
Lhasa terrane (Guillot et al., 2008).

In addition, 25 Ma to 0 Ma is another period considered to contain either the occurrence of slab



breakoff (Jiang et al., 2012; Miller et al., 1999) or lithospheric mantle removal following slab
breakoff (Chung et al., 2005; Nomade et al., 2004). Previous studies suggested that the hotter
asthenosphere considerably raised the geothermal conditions during this period (Chung et al.,
2005; Hou et al., 2004; Williams et al., 2001). Magmatism of the ultrapotassic, shoshonitic, and
calc-alkaline was widespread, which was potentially due to the partial melts of the metasomatized
lithospheric mantle and eclogitized lower crust. An adopted model of convective lithospheric
removal below Lhasa is widely followed (Miller et al., 1999; Molnar et al., 1993; Platt and
England, 1994). The lithospheric removal-related mantle upwelling process has been supported by
geological, geophysical, and petrological studies (Chung et al., 2005; Molnar et al., 1993; Ren and
Shen, 2008; Turner et al., 1993).
Here, slab breakoff and convective lithospheric removal under Tibet are adopted as the
background in this study to discuss the possibility of the delamination process.

**3.  Materials and methods**
**3.1 Starting material**
Natural garnet, omphacite, and epidote samples are collected from eclogite in the Dabie-Sulu
UHPM belt. The compositions of each mineral are determined to be $Prp_{21}Alm_{47}Grs_{31}Sps_1$ (Prp =
pyrope, Alm = almandine, Grs = grossular, and Sps = spessartine) for garnet, $Quad_{48}Jd_{45}Ae_7$
(Quad = Ferrosilite + enstatite + wollastonite, Jd = jadeite, Ae = aegirine) for omphacite, and
$Ca_{2.02}Fe_{0.75}Al_{2.32}Si_{0.16}[SiO_4][Si_2O_7]O(OH)$ for epidote. The compositions of garnet and omphacite
are shown in Figure 2 and are within the range of natural mineral compositions of eclogite from
Tibet. The chemical composition of representative epidote minerals in Tibet shows that the Fe



content of epidote exposed in eclogite is in the range of 0.13-0.25 ($X_{Fe}=Fe^{3+}/(Fe^{3+}+Al^{3+})$)) (Huang
et al., 2015; Li et al., 2017; Liu et al., 2016; Ma et al., 2017; Song et al., 2003; Weller et al., 2016;
Yang et al., 2014), while the Fe content of epidote in this study is 0.24, which is within the Fe
content range of natural epidote. The samples used in this study are representative of garnet,
omphacite, and epidote minerals in natural eclogites from Tibet. The garnet, omphacite, and
epidote with high-quality grains are separated from the eclogite specimens. The above three
samples are crushed into 30×40 $\mu m^2$ chips with a single crystal thickness of 15 μm in our
experiment.
**3.2 Synchrotron X-ray diffraction**
The high-pressure and high-temperature experiment is conducted by a BX90 externally-heated
diamond anvil cell (EHDAC) with ±15° opening angles. The above three single crystals are
loaded into the BX90 EHDAC equipped with a pair of 500 μm culet-size diamond anvils and WC
seats (Figure S1). The rhenium (Re) gasket is pre-indented to a thickness of ~60 μm, and a
cylindrical hole with a diameter of 360 μm is drilled as a sample chamber. Gold powder is also
loaded as the pressure calibrant (Fei et al., 2007), and neon is loaded as the pressure transmitting
medium through the GeoSoilEnviroCARS (GSECARS) gas loading system (Rivers et al., 2008).
The quasi-hydrostatic condition in the sample chamber can be maintained up to ~20 GPa using the
neon pressure transmitting medium (Finkelstein et al., 2017). On the other hand, high temperature
can significantly decrease the deviatoric stress conditions in the sample chamber. Moreover,
previous studies demonstrate that the deviatoric stress disappears at the temperatures of 650 K
with neon as the pressure transmitting medium (Klotz et al., 2009; Meng et al., 1993). Therefore,
the hydrostatic/quasi-hydrostatic conditions can be maintained within the *P-T* range of our





experiment (~700 K, 25 GPa). An automated pressure-driven membrane system is utilized to
generate increasing pressure up to 25.6 GPa. High-temperature conditions up to 700 K are
provided by the heating resistor. Setup details for the employed thermocouples and heaters can be
found in our previous articles (Xu et al., 2019, 2020b; Ye et al., 2021).
*In-situ* synchrotron single-crystal X-ray diffraction (XRD) experiments were performed at
experimental station 13-BM-C of the Advanced Photon Source, Argonne National Laboratory. The
detailed experimental process and associated parameters can be seen in our previous studies (Xu et
al., 2017, 2018, 2020a; Zhang et al., 2017a). The diffraction images and the lattice parameters
were analyzed by the Bruker APEX3 software package (Dera et al., 2013). The specific unit-cell
parameters of the above three samples at each *P-T* condition can be found in Table S1.

**4    Results and discussions**
**4.1 EoS of main minerals for eclogite**
The pressure-volume-temperature (*P-V-T*) data in this study are fitted by the high-temperature
third-order Birch-Murnaghan-EoS (HT-BM3-EoS) (Birch, 1947) to obtain the thermal EoS
parameters including the zero-pressure volume ($V_{T0}$), the isothermal bulk modulus ($K_{T0}$), and its
pressure derivate ($K_{T0}'$) with the following form:
$$P = (3/2) K_{T0} \left[ \left( V_{T0}/V \right)^{7/3} - \left( V_{T0}/V \right)^{5/3} \right] \times \left\{ 1 + (3/4)\left( K_{T0}' - 4 \right)\left[ \left( V_{T0}/V \right)^{2/3} - 1 \right] \right\} \quad (1)$$

where $V_{T0}$ and $K_{T0}$ at different isotherms are expressed by the following equations:
$$V_{T0} = V_0 \, exp \int_{300}^{T} \alpha_T \mathrm{d}T \quad (2)$$

$$K_{T0} = K_0 + \left( \partial K_T / \partial T \right)_P \left( T - 300 \right) \quad (3)$$

where $(\partial K_T/\partial T)_P$ is the temperature derivative of the bulk modulus and $\alpha_T$ ($\alpha_T = \alpha_0 + \alpha_1 T + \alpha_2 T^{-2}$) is
the thermal expansion coefficient at room pressure. Considering the limited high temperature





experimental data in this study, we fixed $\alpha_1$ and $\alpha_2$ as 0 in the fitting.

The thermal EoS parameters are derived using the EoSFit program without any constraints at

high-pressure and room-temperature and high-pressure and high-temperature conditions (Angel et
al., 2014) and are shown in Table S2. Under ambient pressure and temperature conditions, the
measured $V_0$ values of garnet, omphacite, and epidote are 1565.8 (4) Å$^3$, $V_0$ = 423.3 (4) Å$^3$, and V0
= 461.2 (2) Å$^3$, respectively. The fitting parameters under high-pressure and room-temperature
yield $K_{T0}$ = 172 (2) GPa, $K_{T0}'$ = 3.6 (2) for garnet, $K_{T0}$ = 124 (2) GPa, $K_{T0}'$ = 3.7 (4) for omphacite,
and $K_{T0}$ = 122 (1) GPa, $K_{T0}'$ = 2.5 (2) for epidote, respectively.

To evaluate the quality of BM3-EoS fitting in this study, the relationship between the Eulerian

strain ( $f_E = \left[ \left( V_0 / V \right)^{2/3} - 1 \right]$ ) and the normalized pressure ( $F_E = P / \left[ 3 f_E \left( 2 f_E + 1 \right)^{5/2} \right]$ ) of
the main minerals for eclogite is plotted in Figure S2. Linear fitting of the three sets of data
exhibited a negative slope, indicating that the pressure derivative of the bulk modulus ($K_{T0}'$) is less
than 4, which is consistent with our BM3-EoS fittings. The intercept value was obtained by
weighted linear regression of the data points, showing that $F_E(0)$=171 (2) GPa for garnet,
$F_E(0)$=123 (2) for omphacite, and $F_E(0)$=122 (1) for epidote, respectively. The results are
consistent with the fitted isothermal bulk modulus ($K_{T0}$ = 172 (2) GPa for garnet, $K_{T0}$ = 124 (2)
GPa for omphacite, and $K_{T0}$ = 122 (1) GPa for epidote, respectively) within the error range.
Accordingly, the $K_{T0}$ and $K_{T0}'$ obtained by the BM3-EoS fitting are reasonable. Using the $V_0$ fixed
at ambient conditions to fit HT-BM3-EoS, the available EoS parameters, $K_{T0}$ = 171.4 (8) GPa, $K_{T0}'$
= 3.5 (1), $(\partial K_T/\partial T)_P$ = -0.010 (3) GPaK$^{-1}$, and $\alpha_0$ = 2.86 (9) ×10$^{-5}$ K$^{-1}$ for garnet; $K_{T0}$ = 122 (2) GPa,
$K_{T0}'$ = 4.1 (3), $(\partial K_T/\partial T)_P$ = -0.025 (6) GPaK$^{-1}$, and $\alpha_0$ = 4.7 (4) ×10$^{-5}$ K$^{-1}$ for omphacite; and $K_{T0}$ =
122.7 (6) GPa, $K_{T0}'$ = 2.49 (8), $(\partial K_T/\partial T)_P$ = -0.029 (2) GPaK$^{-1}$, and $\alpha_0$ = 4.7 (1) ×10$^{-5}$ K$^{-1}$ for





epidote are derived. The *P-V-T* data fitted through the HT-BM3-EoS model are shown in Figure 3.
**4.2 Comparison with previous studies**
**4.2.1 Garnet**
The thermal EoS parameters of garnet are obtained by fitting the *P-V-T* data to the BM3-EoS. We
compare our results with those of previous studies (Arimoto et al., 2015; Gréaux and Yamada,
2014; Lu et al., 2013; Milani et al., 2015, 2017; Xu et al., 2019; Zou et al., 2012). The $K_{T0}$ of
end-member garnet, pyrope, almandine, grossular, and spessartine crystals is between 158 and 179
GPa, and the bulk modulus of almandine is the largest among the above (Table S3). From Table
S3, it can be seen that the bulk modulus of powder XRD (Arimoto et al., 2015; Gréaux and
Yamada, 2014; Pavese et al., 2001; Zou et al., 2012) are larger than those of single-crystal XRD
(Milani et al., 2015, 2017) with the same composition. The $K_{T0}$ of solid solution garnets (Beyer et
al., 2021; Jiang et al., 2004; Lu et al., 2013; Xu et al., 2019) is also between 158 and 179 GPa
mentioned above and will be affected by the end-member components. The $K_{T0}$=171.4 (8) GPa in
this study is reasonable within this range. The obtained $K_{T0}'$ =3.5 (1) in this study is slightly lower
than that in previous studies. The Eulerian strain and the normalized pressure of the garnet shown
in Figure S2(a) exhibit a negative slope, which indicating $K_{T0}'$ is less than 4. Moreover, compared
with the previous results, the obtained value of $K_{T0}'$ in this study is within the error range
(Supporting Information Text S1). However, there is no obvious correlation between the fitted $K_{T0}$
and $K_{T0}'$ for minerals of different compositions (Fig. S3); hence, the $K_{T0}$ may not be precise when
$K_{T0}'$ is fixed. The value of $(\partial K_T/\partial T)_P$ = -0.010 (3) GPa/K in this study is close to that of Xu et al.
(Xu et al., 2019) obtained through single-crystal XRD experiments, which reflects that the
compositional effect on $(\partial K_T/\partial T)_P$ is minor, but $(\partial K_T/\partial T)_P$ is smaller than that of end-member





garnets obtained from energy-dispersive XRD experiments (Table S3). For the $\alpha_0$, the andradite
has the largest value (3.16 (2)×10$^{-5}$ K$^{-1}$), and the grossular has the smallest value (2.09 (2)×10$^{-5}$ K$^{-1}$)
among the end-member garnets. The thermal expansion coefficient of $Prp_{21}Alm_{47}Grs_{31}Sps_1$ (2.86
(9)×10$^{-5}$ K$^{-1}$) in this study is comparable with previous studies, but the influence of composition
still needs to be considered (Supporting Information Text S2).
**4.2.2 Omphacite**
Many studies have focused on the thermoelastic properties of omphacite (Hao et al., 2019;
Nishihara et al., 2003; Pandolfo et al., 2012b, 2012a; Xu et al., 2019; Zhang et al., 2016) (Table
S3). Most of the results are obtained by the single-crystal XRD method, except for the result of
Nishihara et al. (2003), which was obtained from powder XRD. $K_{T0}'$ shows a higher value of 6.9
(12) in the study of Nishihara et al. (2003), while in others, $K_{T0}'$ is between 4 and 5.7, and the
result of $K_{T0}'$ (4.1) in this study is exactly between the above values. Additionally, according to the
results shown in Table S3, the bulk moduli of omphacite are in the range of 115-123 GPa. In the
study of Xu et al. (2019), an increase in the iron content would decrease $K_{T0}$, and they also
discussed the reasons for the discrepancy in $K_{T0}$ in detail, such as the effective ionic radius,
pressure transmitting medium, and experimental pressure range. Comparing our results with Xu et
al. (2019), we conclude that the incorporation of Fe would reduce the bulk modulus. However,
except for Fe content, there does not seem to be a significant correlation between the other
components and the bulk modulus of omphacite. The $\alpha_0$ of the Di-Jd solid solution is similar (2.64
(2) ×10$^{-5}$ K$^{-1}$-2.8 (3)×10$^{-5}$ K$^{-1}$) but less than that of $Quad_{48}Jd_{45}Ae_7$ (4.7 (4) ×10$^{-5}$ K$^{-1}$) and
$Quad_{53}Jd_{27}Ae_{20}$ (3.4 (4)×10$^{-5}$ K$^{-1}$). It may be inferred that the Ae contents affect thermal expansion.
The $(\partial K_T/\partial T)_P$ of $Quad_{48}Jd_{45}Ae_7$ in this study is -0.025 (6) GPa/K, which is larger than that of

—





$Quad_{57}Jd_{42}Ae_1$ and $Quad_{53}Jd_{27}Ae_{20}$ in the Xu et al. (2019) study.
**4.2.3 Epidote**
The thermal EoS parameters of epidote in this study are compared with those reported in previous
studies (Fan et al., 2014; Gatta et al., 2011; Holland et al., 1996; Li et al., 2020; Qin et al., 2016)
(Table S3). Although the bulk modulus appears to be related to the $Fe^{3+}$ content, it does not show a
good correlation. Increasing the content of $Fe^{3+}$ can enhance the bulk modulus, but the result in
Holland et al. (1996) shows an abnormally large value of 162 (4) GPa, which is much higher than
the 111-133 GPa resulting from other studies. This may be attributed to the fixed $K_{T0}'$ at 4 and
powder XRD methods used in the study of Holland et al. (1996). Furthermore, the $K_{T0}'$ obtained
from powder XRD (Fan et al., 2014; Gatta et al., 2011) is also larger than that from single-crystal
XRD (Qin et al., 2016). The possible reasons for these discrepancies are complicated. Li et al.
(2020) conducted a detailed study on this topic. Previous studies on $\alpha_0$ and $(\partial K_T/\partial T)_P$ of epidote
are limited. The $\alpha_0$ (4.7 (1) $\times 10^{-5}$ $K^{-1}$) in this study is similar to that of Gatta et al. (5.1 (2) $\times 10^{-5}$
$K^{-1}$) (Gatta et al., 2011) and slightly larger than that of Li et al. (3.8 (5) $\times 10^{-5}$ $K^{-1}$) (Li et al., 2020).
In previous studies, only Li et al. (2020) derived the value of $(\partial K_T/\partial T)_P$ (-0.004 (1) GPa/K), which
is much smaller than the absolute value produced in this study (-0.029 (2) GPa/K).

**5    Implications**
In the Himalayan-Tibetan system, lithospheric removal is proposed to occur in either the breakoff
of the subducted slab of the Indian continental lithosphere (Chung et al., 2005; Liu et al., 2014;
Turner et al., 1993; Zhao et al., 2009) or convective removal of the thickened lower part of the
lithosphere (Husson et al., 2014; Miller et al., 1999; Tian et al., 2017; Zhang et al., 2017b). The

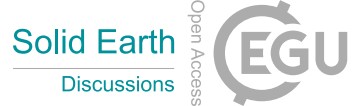

metamorphic eclogitization taking place in the subducted slab and the lowermost crust has been
deduced as the possible cause of subducted slab break-off and the convective removal of the lower
crust (Kind, 2002; Krystopowicz and Currie, 2013; Shi et al., 2015). Increased density in the
eclogitized subducted slab and the lower crust will promote the above two lithospheric removal
modes if the lower crust is weak enough for the negative buoyancy of the mantle lithosphere to be
detached. Therefore, to better consider the role of eclogite density variations in the process of
lithospheric removal, we model the density of minerals and eclogite aggregates along with the
geotherms of Tibet and discuss the effects of the degree of eclogitization on lithospheric removal.

The eclogite chemical data collected in Tibet and examined in our study come from a great

number of eclogite samples collected in previous studies (e.g. Chan et al., 2009; Liu et al., 2019;
Song et al., 2003; Weller et al., 2016; Yang et al., 2009; Zhai et al., 2011a). The eclogite samples
consist of garnet, omphacite, epidote, amphibole, zoisite, symplectite along with minor phengite,
quartz, rutile, and rare apatite, ilmenite, and titanite as accessory minerals. Since the eclogite
samples have suffered retrograde metamorphism, we assume that is largely composed of garnet
and omphacite plus slight epidote before retrograde metamorphism. The accessory phases
observed in natural eclogite are excluded because of their minimal abundance of less than 5%.
Based on the mineral composition data of exposed eclogite in Tibet (Fig. S3) (e.g. Cheng et al.,
2015; Dong et al., 2018; Huang et al., 2015; Jin et al., 2019; Li et al., 2017; Yang et al., 2014; Zhai
et al., 2011b, 2011a), the components of eclogite are 50 vol. % garnet + 45 vol. % omphacite + 5
vol. % epidote (parameterized as a value out of 100) using the normal distribution.

We take into account two different delamination modes, namely, delamination caused by the

separation of the Neo-Tethyan slab (detachment of the subducted Neo-Tethyan oceanic slab) in the



Paleozoic and convective removal of the lower crust of the subducted Indian continent beneath the
Lhasa terrane during the Cenozoic. The temperature and pressure conditions of exposed eclogites
in the Paleozoic and Cenozoic are somewhat consistent with the geothermal lines provided by
previous studies (Fig. 4). The two different delamination modes reflect relatively cold geotherms
and hot geotherms, respectively. Therefore, these geothermal lines are used in our models. The
thermal EoS parameters of eclogitic garnet, omphacite, and epidote are derived through the
HT-BM3-EoS shown in supporting information Table S2.
**5.1 The density of main minerals for eclogite along the geothermal profile in Tibet**
Tibetan eclogite is mainly composed of garnet, and omphacite, with a few epidotes. As shown in
Figure 2, the exposed minerals differ in composition. The specific composition of minerals
constrains the density. Therefore, we refer to the thermoelastic parameters of Xu et al. (2019) and
Nishihara et al. (2003) to depict the density distribution of different components (Fe content) of
garnet and omphacite under Tibetan geothermal lines, respectively. The corresponding
thermoelastic parameters can be seen in Table S3. The mineral compositions of previous studies
are within the range of the Tibetan constituents collected in this study (Fig. 2).
The density distribution of minerals along with relatively cold Tibetan geothermal conditions is
shown in Figure 5 (the results along with hot geotherms can be seen in supporting information Fig.
S6). The result clearly shows that the density of garnet is linked with the iron content. The density
of garnet ($Prp_{21}Alm_{47}Grs_{31}Sps_1$, with 47 mol. % almandine) in this study is higher than that of
low-Fe garnet ($Prp_{28}Alm_{38}Grs_{33}Sps_1$, with 38 mol. % almandine) (Xu et al., 2019) by 2.25% but
lower than that of high-Fe garnet ($Prp_{14}Alm_{62}Grs_{19}Adr_3Sps_2$, with 62 mol. % almandine) (Xu et al.,
2019) by 3.74% at ~80 km (Fig. 5a). With increasing depth, the density of high-Fe garnet





increases by a larger amplitude. This discrepancy may be caused by its smaller degree of thermal
expansion ($2.56 (44) \times 10^{-5}$ K$^{-1}$). Accordingly, the influence of pressure on the density is greater
than that of temperature, which leads to faster increases in density with depth. The density of
omphacite does not show obvious characteristics related to its composition. The density of
omphacite (Quad$_{48}$Jd$_{45}$Ae$_7$, with 7 mol. % aegirine) in this study is lower than that of high-Fe
omphacite (Quad$_{53}$Jd$_{27}$Ae$_{20}$, with 20 mol. % aegirine) (Xu et al., 2019), Quad$_{72}$Jd$_{28}$ (Nishihara et
al., 2003), and Quad$_{57}$Jd$_{42}$Ae$_1$ (with 1 mol. % aegirine) (Xu et al., 2019) by 2.07%, 1.63%, and
0.99%, respectively, at ~80 km (Fig. 5b). The presence of iron in certain quantities does increase
the density of omphacite, but the density of omphacite is also affected by other elements, such as
calcium and magnesium. Moreover, thermal EoS parameters are also of vital importance to
calculate the density. The relatively low thermal expansion of Quad$_{72}$Jd$_{28}$ ($2.7 (3) \times 10^{-5}$ K$^{-1}$) and
Quad$_{57}$Jd$_{42}$Ae$_1$ (with 1 mol. % aegirine) ($2.8 (3) \times 10^{-5}$ K$^{-1}$) may further enhance the increasing rate
of density with depth. It is worth noting that the densities of Quad$_{48}$Jd$_{45}$Ae$_7$ (with 7 mol. %
aegirine) in this study and Quad$_{57}$Jd$_{42}$Ae$_1$ (with 1 mol. % aegirine) of Xu et al. (2019) are the same
under ambient conditions but inconsistent under high-pressure and high-temperature conditions.
Therefore, the $K_{T0}$ and $K_{T0}'$ of the two omphacites are somewhat consistent with each other, while
the thermal expansion and $(\partial K_T / \partial T)_P$ are different. Collectively, the thermal EoS parameters are of
the essence in the derivation of the mineral density.
**5.2 The density of eclogite in Tibet**
Eclogitized crust and lithospheric mantle may be potential factors causing delamination (Faccenda
et al., 2009; Krystopowicz and Currie, 2013; Ueda et al., 2012). The density of eclogite and
peridotite can provide new constraints to control the breakoff of the subducted slab and convective



removal of the lithosphere in the process of delamination. Therefore, we plot the density
distribution of eclogite with different garnet contents and peridotite along the Paleozoic and
Cenozoic Tibetan geotherms, as shown in Figure 6. In our model, the mineral composition of
Tibetan eclogite is in the range of 40 vol. % garnet + 55 vol. % omphacite + 5 vol. % epidote to 60
vol. % garnet + 35 vol. % omphacite + 5 vol. % epidote based on the exposed eclogite in Tibet
(the composition of epidote is only 5 vol. % default due to its low content in this study). The
composition of surrounding peridotite consists of 70 vol. % olivine + 25 vol. % orthopyroxene + 3
vol. % clinopyroxene + 2 vol. % spinel (Konstantinovskaia et al., 2003; Yang et al., 2019; Zhao et
al., 2021). The densities of eclogite and peridotite aggregates are obtained considering their
arithmetic mean. The density of each mineral under a specific temperature and room pressure can
be calculated by the following equations:

$$\rho(T,0) = \rho_0 \exp\left[ -\int_{T_0}^{T'} \alpha(T)\,\mathrm{dT} \right] \tag{1}$$

$$\alpha(T) = \alpha_0 + \alpha_1 T + \alpha_2 T^{-1} + \alpha_3 T^{-2} \tag{2}$$

where $\rho$ (T, 0) and $\rho_0$ are the densities of specific and ambient temperatures, respectively. $\alpha$ (T) is
the thermal expansion coefficient, which is a function of temperature. Here, we define $\alpha$ (T) to be
a constant (Table S2). The relationship with pressure is obtained according to the third-order
Birch-Murnaghan equation of state and Euler finite strain theory (Birch, 1947, 1978):

$$P = (3/2)\left(K_0 + (T-T_0)(\partial K/\partial T)_P\right)\left[(V_{T0}/V)^{7/3} - (V_{T0}/V)^{5/3}\right]\left\{1 + (3/4)(K_{T0}'-4)\left[(V_{T0}/V)^{2/3} - 1\right]\right\}$$

(3)

where $V_{T0}$, $K_{T0}$, and $K_{T0}'$ are the unit cell volume, bulk modulus, and its pressure derivative,
respectively, $V$ is the unit cell volume at high pressures, and $(\partial K_T/\partial T)_P$ is the temperature
derivative of the bulk modulus. The densities of each mineral under specific temperature and





pressure conditions are derived by the following formula:

$$\rho(T,P) = \rho(T,0)V(P,T) \qquad (4)$$

Most changes in the deep conditions of the Earth are progressing slowly, so there is adequate time
for recrystallization to relieve the maximum stress point (Robertson, 1988; Skinner, 1966). Here,
we assume that the elastic-plastic interaction among different minerals and possible deviations
from hydrostatic conditions are ignored and the density of the eclogite aggregate can be obtained
by the arithmetic mean as follows:

$$\overline{\rho} = \sum \lambda_i \rho_i(T,P) \qquad (5)$$

where the subscript $i$ denotes the $i$th mineral of the upper mantle, and $\lambda$ is the proportion of each
mineral.
The densities of Tibetan eclogite (with the garnet composition of $Prp_{21}Alm_{47}Grs_{31}Sps_1$, the
omphacite composition of $Quad_{48}Jd_{45}Ae_7$, and the epidote composition of
$Ca_{2.02}Fe_{0.75}Al_{2.32}Si_{0.16}(SiO_4)(Si_2O_7)O(OH))$ and peridotite (with the olivine composition of
$Fo_{89.9}Fa_{10.1}$, the orthopyroxene composition of $En_{89.6}Fs_{9.7}Wo_{0.7}$, the clinopyroxene composition of
$Quad_{88.5}Jd_{11.5}$, and the spinel composition of
$(Mg_{0.790}Fe_{0.204}Ni_{0.005}Ti_{0.001})_{1.000}(Al_{0.821}Cr_{0.158}Fe_{0.021})_{2.002}O_4)$ in this study along the Paleozoic
geothermal line are shown in Figure 6a. The results show that the increase in garnet has a
profound influence on the density of eclogite. For every 10% increase in garnet, the density of
eclogite increases by ~1.7%. The garnet content in Tibetan eclogite is estimated to be 40 vol. %-60
vol. % (Fig. S7). The densities of this part of eclogite are 3.55-3.67 $g/cm^3$, which is approximately
7.7%-11.5% more than that of peridotite (3.29 $g/cm^3$) at ~80 km. The density difference between
eclogite and peridotite is 0.25 $g/cm^3$-0.38 $g/cm^3$ (Fig. 6b). At the same time, we also consider the



density of eclogite aggregates without epidote (Fig. S7). The results show that 5 vol. % epidote
has little effect on the density of eclogite, especially eclogite with garnet contents of 50 vol. %-60
vol. %. To account for the role of iron, the density distributions of high-Fe
($Prp_{14}Alm_{62}Grs_{19}Adr_3Sps_2$ and $Quad_{53}Jd_{27}Ae_{20}$ and low-Fe eclogite ($Prp_{28}Alm_{38}Grs_{33}Sps_1$ and
$Quad_{57}Jd_{42}Ae_1$) are plotted to better constrain the range of eclogite density (Fig. S9) (Xu et al.,
2019). For high-Fe and low-Fe eclogites, the densities of eclogite increase by ~1.9% and ~1.4%
for each 10% increase in garnet, respectively. The densities of eclogite are 3.64 $g/cm^3$-3.78 $g/cm^3$
for high-Fe content and 3.53 $g/cm^3$-3.63 $g/cm^3$ for low-Fe content at ~80 km. Furthermore, the
densities of high-Fe and low-Fe eclogites are 10.6%-14.9% and 7.2%-10.3% higher than the
surrounding peridotite, respectively. For a more straightforward comparison, taking eclogite
containing 50 vol. % garnet as an example (Fig. S10), the densities of high-Fe eclogite, low-Fe
eclogite, and Tibetan eclogite at ~80 km are 3.71 $g/cm^3$, 3.58 $g/cm^3$, and 3.61 $g/cm^3$, respectively.
An increase in the iron content can substantially increase the density of eclogite, although it will
be constrained by the thermal EoS parameters of minerals.
Similarly, we also discuss the density profile along the Cenozoic geothermal line, which can be
seen in supporting information Text S3. In any case, the density difference caused by eclogite may
be one of the prominent factors instigating the delamination process.
**5.3 Influence of the degree of eclogitization on the density of the subducted slab**
Eclogite in the mantle, which is believed to be 5%-10% denser than peridotite (Garber et al.,
2018), is responsible for the excess compositional density. Furthermore, some calculations
propose that the degree of eclogitization of the subducted slab is a key factor in the delamination
process (Matchette-Downes et al., 2019). To investigate the influence of the degree of



eclogitization in the delamination process, we plot the density variations with different mineral
compositions under different degrees of eclogitization (Fig. 7). We consider eclogitization in the
lithospheric mantle of the subducted slab. In our preferred model, the 7-km thick subducted
oceanic crust becomes eclogite, while the lithospheric mantle constrains a different amount of
eclogite. Since the subducted Indian oceanic slab might be fragmented into several pieces, the
longitudinal size of the fractured slab is postulated to be 60 km (Peng et al., 2016). Our estimated
average density of the fragmented slab with various degrees of eclogitization is shown in Figure
7a. The results clearly show that the density increases monotonically with the garnet content and
the degree of eclogitization. The garnet content is of profound importance to the density of
eclogite. The higher the proportion of garnet is, the greater the density increases with increasing
degrees of eclogitization. The garnet content in Tibetan eclogite is estimated to be between 40-60
vol. %. Taking garnet with an average volume percentage of 50 vol. % in Tibetan eclogite as an
example, the density of eclogitized subducted slabs ranges from 3.37 g/cm$^3$ with 10%
eclogitization to 3.61 g/cm$^3$ with 100 vol. % eclogitization. For a garnet content of 50 vol.%, the
density increases by 0.026 g/cm$^3$ per 10 vol. % increase in the degree of eclogitization. The
density will increase with increasing garnet contents, from 0.004 g/cm$^3$ for 10 vol. % to 0.048
g/cm$^3$ for 90 vol. %. The densities of high-Fe and low-Fe eclogitized fragmented slabs are also
shown in Figure S11. The high-Fe content shows that the density variation increases with the
degree of eclogitization from 0.007 g/cm$^3$ for 10 vol. % to 0.064 g/cm$^3$ for 90 vol. % garnet, while
the low-Fe content shows a density change from 0.004 g/cm$^3$ for 10 vol. % garnet to 0.045 g/cm$^3$
for 90 vol. % garnet.
**5.4 Delamination in Tibet**



The development of delamination is associated with the instability of the lower crust and the
mantle lithosphere. The eclogitization of the subducted slab and lower crust plays a vital role in
the process of delamination due to the high density of eclogite (Anderson, 2005; Lee et al., 2011),
which makes the formation denser than the surrounding mantle lithosphere and provides critical
negative buoyancy (Göğüş and Ueda, 2018; Krystopowicz and Currie, 2013). The densities of the
eclogitic lower crust and mantle lithosphere during slab subduction and convective removal are
sufficiently higher than that of the asthenosphere and are good candidates for the initiation of
destabilization.
**5.4.1 Subducted slab breakoff**
A series of collisional breakoff events is proposed to have occurred throughout 60-45 Ma in Tibet
(Chung et al., 2005, 2009; Ma et al., 2014; Zhu et al., 2015). The formation of eclogite
presumably kick-starts slab breakoff during the subduction of the Indian oceanic plate underthrust
below the southern margin of Tibet. The subducted Indian oceanic slab fragmented into several
pieces, due to what has been identified as a high-velocity anomaly (Peng et al., 2016; Razi et al.,
2014; Shi et al., 2020b). The seismological evidence of high density (Hetényi et al., 2007), high $V_P$
(Schulte-Pelkum et al., 2005), and low longitudinal/transverse ($V_P/V_S$) ratios (Wittlinger et al.,
2009) further confirms that there may be variable degrees of eclogitization beneath Tibet. Figure 7
shows the density profile of subducted slabs with different garnet compositions, different degrees
of eclogitization, and variable densities compared with the surrounding peridotite. An increasing
degree of eclogitization and an enhanced garnet content in eclogite increases the density difference
between the slab and the surrounding peridotite. Previous studies have made preliminary estimates
of the average density from the isostatic balance and geoid anomalies, and postulated that the

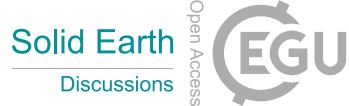

density excess could be between 0-0.19 g/cm$^3$ (Matchette-Downes et al., 2019). For Tibetan
eclogite containing 40 vol. %-60 vol. % garnet, if the lithospheric mantle is a mixture of peridotite
and eclogite with a density anomaly of 0.19 g/cm$^3$, our model requires a range of 37%-64%
degrees of eclogitization. If the eclogite is high-Fe, only a 30%-48% degree of eclogitization is
needed to produce the density difference (Fig. S11), while an eclogitization degree is in the range
of 49%-74% is needed for the low-Fe eclogite. However, some seismological data show that the
crust or lithospheric mantle being only ~30% eclogitized might cause gravitational instability in
Tibet (Matchette-Downes et al., 2019; Shi et al., 2020a), which is lower than our estimation. Our
results clearly show that density excess is closely linked with garnet content and eclogitization
degree. If eclogite has a high garnet content, a relatively low degree of eclogitization could
instigate the delamination of slab breakoff.
On the other hand, the presence of a weak lower crust and a vertical conduit to accommodate
asthenosphere influx is also necessary for the delamination process. The weak layer between the
residual crustal and downward peeling lithosphere layer (and/or lower crust) (Göğüş and Ueda,
2018) could promote the initiation and propagation of delamination. Therefore, very high
temperatures and relatively low lower-crustal viscosities are also other controlling factors of
delamination (Göğüş and Pysklywec, 2008; Morency, 2004; Valera et al., 2011). Here, we assume
that the length of the fractured slab is 60 km, which drops 80 km over 45 Ma and that the viscosity
of the asthenosphere is $5*10^{20}$ Pa·S (Wang et al., 2019). By using Stokes' Law (Supporting
Information Text S4), ignoring the thermal disturbance, and assuming the most ideal conditions,
the density difference caused by eclogite needs to be at least 0.14 g/cm$^3$ to produce such
delamination. The result is close to those discussed above in gravity anomalies.



In particular, the presence of eclogite with a greater abundance of garnet, a higher-Fe content,
and a greater degree of eclogitization would instigate the delamination process of slab breakoff.
**5.4.2 Removal of the eclogitized lower crust**
The thickened lower crust undergoes "convective removal" due to gravitational instability, which
is another type of delamination that occurred in Tibet from 25 Ma to 0 Ma (Chung et al., 2005;
Nomade et al., 2004). The convective removal of the lithosphere during delamination corresponds
to higher temperature conditions (Craig et al., 2020). In this circumstance, the density of Tibetan
eclogite is 7.6%-11.6% denser than the surrounding peridotite at ~60 km (Fig. 6b), which is
analogous to the results in the case of subducted slab detachment. This result is also in ample
agreement with the result obtained by Garber et al. (2018), which noted that eclogite is 5%-10%
denser than peridotite. The density difference between eclogite and peridotite is 0.24 $g/cm^3$-0.37
$g/cm^3$ with 40 vol. %-60 vol. % garnet in Tibet (Fig. 6d). During this stage, it is believed that
delamination of the thickened, eclogitized lower crust has occurred. Similarly, Stokes' law can be
used considering ideal conditions without any thermal disturbance. If the falling block is assumed
to be approximately 30 km in the longitudinal direction and the viscosity of the asthenosphere is
$5*10^{20}$ Pa·S, the falling block can drop by 75-115 km within 25 Ma. For eclogite with a high-Fe
content, a density difference of 0.35 $g/cm^3$-0.50 $g/cm^3$ makes the fragmented block capable of
falling 105-155 km, while the density difference of 0.24 $g/cm^3$-0.33 $g/cm^3$ with a low-Fe content
makes the block able to fall 75-102 km (Fig. S12). The fragmented block with a high-Fe content
can fall a larger distance at the same time, indicating that the high-Fe content is more likely to
promote the occurrence of delamination. This result is consistent with the high-velocity
anomalous blocks identified at 100-200 km by seismic tomography (Peng et al., 2016; Shi et al.,



2016, 2020a).
In summary, density contrasts can provide a stimulus for the initiation of instability. It is
accepted that eclogite with a high garnet content and a high Fe content and a high proportion of
eclogite in the lithospheric mantle may have strongly promoted delamination during the process of
India-Asia collision from the perspective of density.

**6. Conclusion**
The *P-V-T* EoS of the main minerals of eclogite is combined with its mineral composition
and the geothermal line to derive the density of Tibetan eclogite in this study. We offer a new
perspective by obtaining the thermal EoS for the main minerals of eclogite in a single experiment.
The thermal EoS parameters of the main minerals of eclogite are derived by fitting the *P-V-T* data
to the HT-BM-EoS. The density of minerals along the Tibetan geotherm shows that the density is
closely related to its composition and thermal EoS parameters. Increasing iron contents increase
the density of minerals, but if the molecular masses of two minerals are similar, the thermal EoS
parameters play a pivotal role. The garnet content profoundly increases the density of eclogite. For
every 10 vol. % increase in garnet, the density of eclogite increases by approximately 1.7%. The
density of Tibetan eclogite is approximately 7-11% denser than that of the surrounding peridotite.
An increasing proportion of garnet, Fe content, and degree of eclogitization enhance the density
difference to facilitate the delamination process. For Tibetan eclogite containing 40-60 vol. %
garnet, 37-64% degrees of eclogitization can produce the same density difference as obtained by
the isostatic balance and the geoid anomaly. According to a rough calculation, the fragmented
block will fall 75-155 km. A high-Fe content is more likely to promote delamination. Eclogite is a



531 good candidate for the initiation of instability and may be more susceptible to inducing the

532 breakoff of the subducted slab or the gravitational removal of the lower crust during the process of

533 the India-Asia collision.


535 **Data availability**

536 All the data presented in this paper are available upon request.


538 **Author contributions**

539 All authors contributed to the preparation and revision of the manuscript. Z. Ye: Data curation,

540 Investigation, Formal analysis, Writing-original draft, Writing-review & editing. D. Fan:

541 Investigation, Conceptualization, Supervision, Methodology, Funding acquisition, Writing-review

542 & editing. B. Li: Data curation, Writing-review & editing. Q. Tang: Software, Validation,

543 Writing-review & editing. J. Xu: Investigation, Supervision, Writing-review & editing. D. Zhang:

544 Formal analysis, Writing-review & editing. W. Zhou: Investigation, Conceptualization,

545 Supervision, Writing-review & editing.


547 **Competing interests**

548 The authors declare that they have no conflict of interest.


550 **Disclaimer**

551 Publisher's note: Copernicus Publications remains neutral with regard to jurisdictional claims in

552 published maps and institutional affiliations.




**Acknowledgments**
This project was funded by the National Natural Science Foundation of China (Grant Nos.
42172048, U2032118 and 41802043), the Youth Innovation Promotion Association CAS (Dawei
Fan, 2018434), the Chinese Academy of Sciences "Light of West China" Program (2019), the
Guizhou Provincial Science and Technology Projects (QKHJC-ZK[2021]ZD042), and the
Innovation and Entrepreneurship Funding of High-Level Overseas Talents of Guizhou Province
(Dawei Fan, [2019] 10).

**Supplementary Information.** The supplementary information describes the density profile of
garnet at high temperature, density profile along Cenozoic geothermal line, data of unit-cell
parameters of eclogite minerals, thermal EoS parameters of this study and previous researches,
figures of Eulerian finite stain-normalized pressure ($F_E$ -$f_E$), isothermal bulk modulus ($K_{T0}$) and its
pressure derivative ($K_{T0}'$) plot of garnet and omphacite, normal distribution of eclogite minerals,
density evolution of minerals, and density profile of different Fe-content eclogite.

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

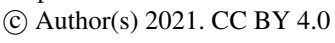

**Figure:**

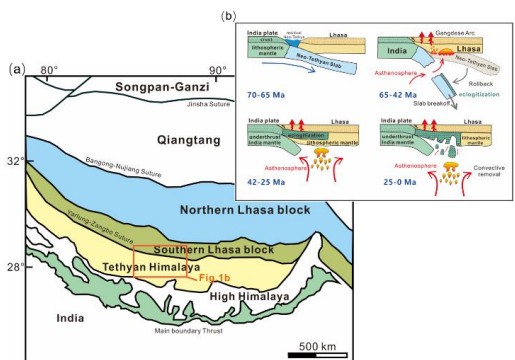


**Figure 1.** (a) Schematic geological map of the Tibetan Plateau (modified from Chung et al. 2005
and Wang et al. 2010). (b) Interpretive geological cartoon of India-Asia collision evolution . 70-65
Ma: The flat Neo-Tethyan oceanic slab subducts beneath Tibet with the closure of the Neo-Tethys
Ocean. 65-42 Ma: The rollback of the Neo-Tethyan slab breaks off after densification by
eclogitization. 42-25 Ma: The subduction of the Indian continent continued at a low subduction
angle beneath the Lhasa terrane and was accompanied by heavy thermal perturbation. 25-0 Ma:
The thickened eclogitic lower crust undergoes the "convective removal" of delamination due to
gravitational instability.



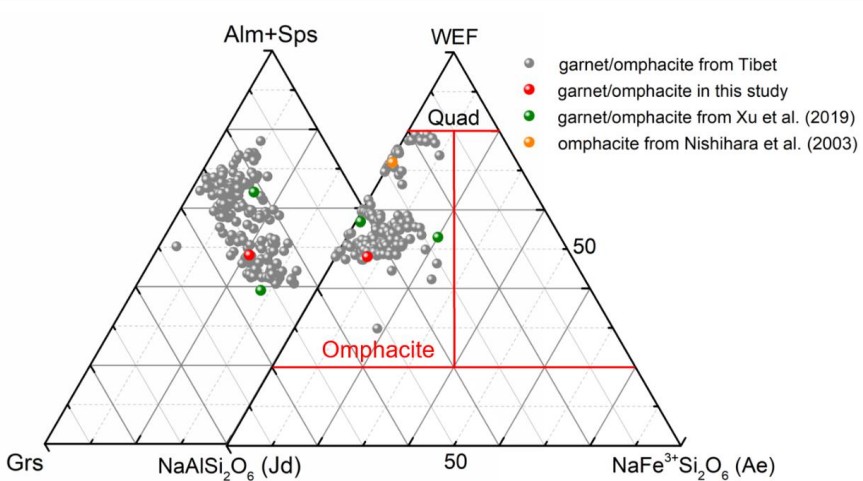


**Figure 2.** Composition of garnet and omphacite in eclogites from Tibet and this study. The gray

solid circles represent the components of garnet and omphacite collected from previous studies in

Tibet (e.g. Chan et al., 2009; Liu et al., 2019; Song et al., 2003; Weller et al., 2016; Yang et al.,

2009; Zhai et al., 2011a). The green solid circles are garnet and omphacite with different Fe

contents according to Xu et al. (2019). The orange solid circles are omphacite according to

Nishihara et al. (2003). The red solid circles are the components of garnet and omphacite in this

study. Prp = pyrope, Alm = almandine, Grs = grossular, Sps = spessartine, Quad = Ferrosilite +

enstatite + wollastonite, Jd = jadeite, and Ae = aegirine.


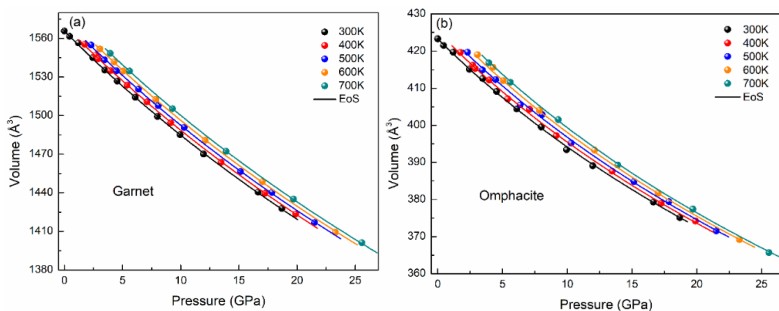




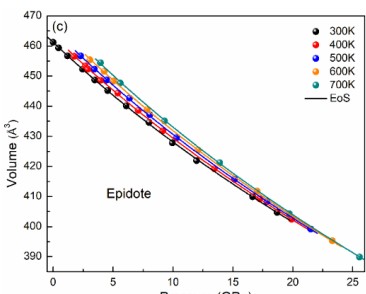


**Figure 3.** Pressure-volume-temperature relations of garnet (a), omphacite (b), and epidote (c).
Isothermal compression curves are calculated by using the thermoelastic parameters obtained in
this study.

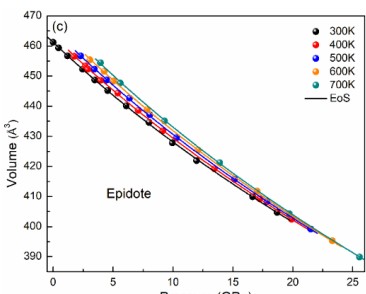


**Figure 4.** The geothermal lines of Tibet. The black line from Wang et al. (2013) represents the
Paleozoic geotherm, which indicates relatively cold conditions. The delamination of the
Neo-Tethyan slab breakoff accrues under such conditions. The orange line represents the
geothermal line of surrounding Tibet during this period (Nábělek and Nábělek, 2014). The blue
line represents the Cenozoic geotherm in Tibet (Craig et al., 2020), which indicates a relatively hot
geotherm under the situation of convective removal. The red line represents the geothermal line of
the surrounding Tibet in this situation (Wang et al., 2013). The green line is the adiabatic line. The
black and blue solid circles correspond to the temperature and pressure conditions of exposed
Tibetan eclogite in the Paleozoic and Cenozoic, respectively. The Paleozoic samples are referred



from (Cheng et al., 2012, 2015; Dong et al., 2016, 2018; Huang et al., 2015; Li et al., 2017; Liu et
al., 2019; Tang et al., 2020; Weller et al., 2016; Yang et al., 2019, 2014) and the Cenozoic samples
are referred from (Chan et al., 2009; Corrie et al., 2010; Hacker, 2000).

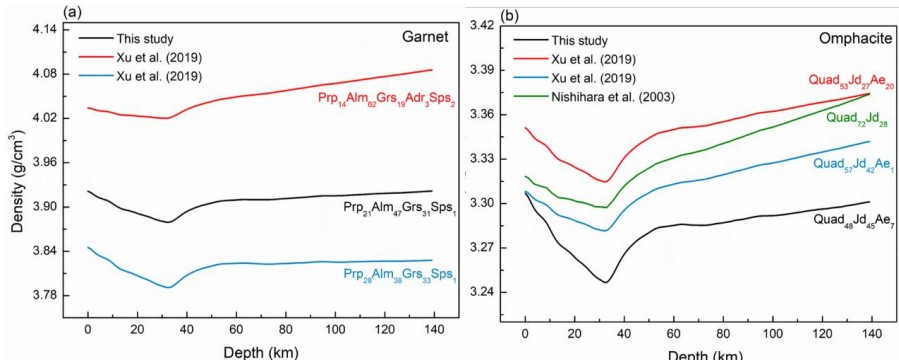


**Figure 5.** Density profiles of garnet (a) and omphacite (b) along with the cold Tibetan geothermal
line (Wang et al., 2013). The garnets of $Prp_{21}Alm_{47}Grs_{31}Sps_1$ and $Prp_{28}Alm_{38}Grs_{33}Sps_1$ are from
Xu et al. (2019). The omphacites of $Quad_{53}Jd_{27}Ae_{20}$ and $Quad_{57}Jd_{42}Ae_1$ are from Xu et al. (2019)
and $Quad_{72}Jd_{28}$ is from Nishihara et al. (2003).

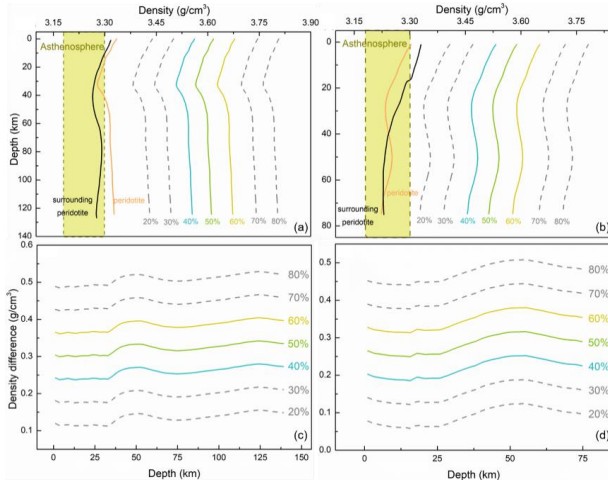




**Figure 6.** Density profiles of eclogite and peridotite assemblages ((a) and (b)) and density

difference between eclogite and peridotite ((c) and (d)) in Tibet along the Paleozoic and Cenozoic

geothermal lines under the conditions of Neo-Tethyan oceanic slab detachment (a) (Wang et al.,

2013) and subduction of the Indian continental margin beneath the Lhasa terrane (b) (Craig et al.,

2020). The percentage represents the content of garnet in eclogite, of which epidote accounts for 5

vol. % by default. The orange curve and black curve show the density profile of peridotite with a

composition of 70 vol. % olivine + 25 vol. % orthopyroxene + 3 vol. % clinopyroxene + 2 vol. %

spinel. The orange line shows the density of peridotite in the lithospheric mantle along the

Paleozoic (a) (Wang et al., 2013) and Cenozoic (b) (Craig et al., 2020) geothermal lines, and the

black curve indicates that the density of peridotite in the surrounding lithospheric mantle is along

the Paleozoic (a) (Nábělek and Nábělek, 2014) and Cenozoic (b) (Wang et al., 2013) geothermal

lines in Tibet. The shaded region is the density range of the asthenosphere (Chen and Tenzer, 2019;

Levin, 2006; Panza et al., 2020; Singh and Mahatsente, 2020).

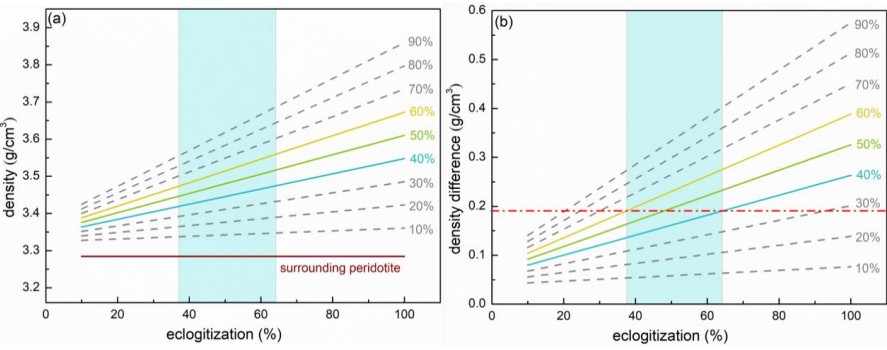

**Figure 7.** (a) The effect of eclogitization on the density of the subducted slab at ~80 km (2.6 GPa

and 625 °C) along the Paleozoic geothermal line. The percentage on the right represents the

content of garnet and the content of epidote is fixed at 5 vol. % by default. The content of garnets



in Tibet is between 40 vol. % and 60 vol. %. The density represents the average density of the
subducted slab with the entire eclogitic ocean lower crust and partially eclogitized lithospheric
mantle, where the degree of eclogitization refers to the lithospheric mantle. The rufous line
represents the average density of surrounding peridotite in this study. The blue shading indicates
the possible degree of eclogitization. (b) Density difference between eclogite with different
degrees of eclogitization and surrounding peridotite. The red dashed solid line represents a density
excess of 0.19 g/cm$^3$ from the isostatic balance and the geoid anomaly (Matchette-Downes et al.,

1076    2019).
