# Peer review of "Thermal equation of state of the main minerals of eclogite: Constraining the density evolution of eclogite during delamination process in Tibet Zhilin Ye a,b, Dawei Fan a,\*, Bo Li c, Qizhe Tang d, Jingui Xu e,\*, Dongzhou Zhang e, Wenge Zhou a a Key Laboratory of High-Temperature and High-Pressure Study of the Earth's Interior, Institute of Geochemistry, Chinese Academy of Sciences, Guiyang, Guiz"

_Solid Earth, 2021_

## Author Comment (AC1)

**Reviewer 1**

*Review of Delamination in Tibet: deriving constraints from the density of eclogite by Ye et al.*

*General comments*

*This paper established new equations of state for constituent minerals of eclogite and discuss the density of these minerals and eclogite in the mantle depth with implications for possible role of eclogite density in delamination of lower crusts and lithospheric mantle.*

*High-pressure and temperature experiments seem to have been done nicely in externally heated diamond cell to 700 K. However the experimental temperatures are low compared to the Debye temperatures of the minerals and its applicability to mantle temperatures needs some attention. The choice of the starting material needs clarification. Some of the wordings are not very clear to me, for example, eclogitization of lithospheric mantle does not make sense to me. Some equations are incorrect and wrongly used. I don't think the title represents what the authors studied here, as they did not discuss the mechanism of delamination in Tibet as a function of the density of eclogite; what they did is construction of EoS of the minerals. Therefore, I'm afraid that the current version is not at the publishable level.*

Enclosed, please find our replies to the reviewer's comments in our revised manuscript entitled "Delamination in Tibet: deriving constraints from the density of eclogite".

We want to appreciate the reviewer for the thoughtful and thorough comments that have resulted in a substantially improved in the revised version of our manuscript. Point-by-point responses to the reviewer's comments are shown in detail below. We have also indicated our changes are marked in red in the revised manuscript. We believe that we have addressed all the reviewer's concerns adequately.

In the following, the reviewer's comments are shown in boldface and are followed by our replies in normal text. References cited are provided at the end of the response letter.

**Specific comments:**
*1. Comment:"The title does not represent what is done in this paper. The authors established EoS of minerals, calculated the density of eclogite and peridotite, and compared them. The only discussion they made regarding delamination processes is the percentage of eclogitizaiton when the rock density reaches the value for isostatic balance. I do not understand how this puts constraints on the delamination processes."*
**Reply:** Thank you for your comment. The purpose of this paper is to emphasize that eclogite is an important factor in the occurrence of delamination and how does the density of eclogite affect the delamination process. The reviewer's suggestion is constructive. We changed our title to: Thermal equation of state of the main minerals of eclogite: Constraining the density evolution of eclogite during delamination process in Tibet.
*2. Comment:"Why did they use natural samples as the starting material? As the authors said themselves (L300-302), natural samples may not represent what they were in the mantle due to retrograde metamorphism."*

**Reply:** Thank you for your comment. The initial samples used in this study are natural rock-forming minerals of eclogite. By summarizing the composition of exposed eclogititic minerals (Figure 1), we found that the composition of minerals used in this study was within the range of natural rock-forming mineral compositions of eclogite from Tibet. This part was also explained in the Line 152-158. Hence, the natural samples used as the starting material is reasonable. In the Line 300-302, we would like to emphasize that exposed eclogites are usually undergo retrograde metamorphism, and the mineral content does not accurately represent the composition of eclogite. Therefore, we corrected the composition of pre-retrograde eclogites by the content of other minerals.

[Figure]

**Figure 1**. Composition of garnet and omphacite in eclogites from Tibet and this study.

*3. Comment :"The thermal EoS was formulated using the high-T Birch-Murnaghan formulation. This is an empirical formulation to describe the high-T behaviour and an extrapolation to the mantle temperature needs attention. Why did not the authors use a thermal pressure model such as using the Mie-Gruneisen relation? The authors should check the high-T behaviour of their EoS by calculating the thermal expansivity at high pressure; it should be increasing with temperature (which will be levelling off above the Debye T). To my eye, omphacite and epidote EoS seem to violate this rule at about 20 GPa, namely the thermal expansivity decreases with T."*

**Reply:** Thank you for your constructive comments. We tried to use the Mie-Gruneisen-Debye thermal-pressure EoS to fit our high-temperature and high-pressure data. Due to the need for some additional parameters and many assumptions required by this model, we can not obtain more appropriate values than using high-T Birch-Murnaghan formulation. Accordingly, we chose another thermal-pressure model Holland-Powell EoS to re-fit our data. The fitting result is shown in Figure 2. The high-T behaviors of omphacite and epidote using this thermal-pressure EoS show more accurate results. The thermal EoS parameters derived from BM3-HP (third-order Birch-Murnaghan compressional EoS in combination with the Holland-Powell thermal-pressure EoS) compared with BM3-Isothermal (third-order Birch-Murnaghan compressional EoS in combination with the Fei-type model) is shown in Table 1, especially the big difference in thermal expansion. From the thermoelastic parameters and the fitted curve, it is apparent that the BM3-HP equation is more suitable for fitting the high-pressure and high-temperature data in this study. We recalculated the

density of the minerals and eclogite through the newly obtained thermoelastic parameters, and made changes in the corresponding figures and texts marked in red.

**Table 1.** Thermal EoS parameters derived from the fitting of P-V-T data to the BM3-Isothermal EoS and BM3-Holland-Powell thermal pressure EoS.

| | | BM3-Isothermal | BM3-HP |
|---|---|---|---|
| Garnet | $V_0$ (Å$^3$) | 1565.8(4) | 1566.05(25) |
| | $K_{T0}$ (GPa) | 171.4(8) | 170.0(1.3) |
| | $K_{T'0}$ | 3.5(1) | 3.82(14) |
| | $\alpha_0$ (10$^{-5}$ K$^{-1}$) | 2.86(9) | 2.71(5) |
| Omphacite | $V_0$ (Å$^3$) | 423.3(4) | 423.48(24) |
| | $K_{T0}$ (GPa) | 122(2) | 121.4(2.9) |
| | $K_{T'0}$ | 4.1(3) | 3.97(34) |
| | $\alpha_0$ (10$^{-5}$ K$^{-1}$) | 4.1(3) | 3.73(20) |
| Epidote | $V_0$ (Å$^3$) | 461.2(2) | 461.57(23) |
| | $K_{T0}$ (GPa) | 122.7(6) | 123.8(1.8) |
| | $K_{T'0}$ | 2.49(8) | 2.04(15) |
| | $\alpha_0$ (10$^{-5}$ K$^{-1}$) | 4.7(1) | 3.04(13) |

[Figure]

**Figure 2.** Pressure-volume-temperature relations of garnet (a), omphacite (b), and epidote (c). Isothermal compression curves are calculated by using the thermoelastic parameters obtained in this study.

**4. Comment:** *"Density is an intensive property and the simple average scheme cannot be applied to obtain the rock density."*

**Reply:** Thank you for your comment. Density in this study is obtained by volume percentage rather than mole percentage. Volume is an extensive property, so the calculated mass is also an extensive property. Furthermore, we assume an ideal condition without taking into account any elastic-plastic interactions between different minerals and any possible deviations from hydrostatic conditions. Most changes of conditions in the Earth proceed slowly, so there is ample time for crystallization to relieve the points of greatest stress and close the pores. Although all rocks have some porosity, they can be largely eliminated by about 1000 atmospheres of pressure, indicates that such readjustments and compensations can be ignored (Robertson, 1988; Skinner, 1966). Moreover, this density calculation profile was also used in previous studies (Faccincani et al., 2021; Ganguly et al., 2009; Ohtani and Maeda, 2001). In the light of these facts, it is believed that the density of any rock under the deep Earth conditions can be obtained by averaging the weighted density of its constituent minerals.

**5. Comment:** *"Equations 1 and 4 are wrong. Not sure if these are typos, or the authors calculated those properties using these equations. If they did, all the calculations were incorrectly done."*

**Reply:** Thank you for your comment. We feel sorry for some typos due to our negligence. The correct equations are shown as below:

Equation 1: $\rho(T',0) = \rho_0 \exp\left[-\int_{T_0}^{T'} \alpha(T) dT\right]$

Equation 4: $\rho(T,P) = \rho(T,0)\dfrac{V(T,0)}{V(T,P)}$

For a more detailed description, we modified the equation 4 into the following form:

$$\rho(T,P) = \frac{V(T,0)}{V(T,P)} \times \frac{Z \times M}{N_a \times V_0}$$

**6. Equation 2 is not needed here as the authors dropped off the higher terms."**

**Reply:** We agree with the reviewer's comment. Equation 2 here is just to illustrate the expression used for thermal expansion and we did define the thermal expansion of minerals as a constant. We deleted the Equation 2 here.

**7. I do not understand what it means by eclogitization of lithospheric mantle. Does this mean mantle peridotite is getting eclogite?"**

**Reply:** Thank you for your comment. The eclogitization here refers to the amount of eclogite in the lithospheric mantle. Previous studies indicate that eclogites in peridotite can originate either by direct HP crystallization of omphacite and garnet from transient melts in the mantle, or by prograde HP recrystallization of plagioclase–bearing basic dikes in peridotite (Medaris et al., 2018). To be more precise, we revised our expressions in the revised manuscript at Line 421-424: "We consider eclogitization in the lithospheric mantle of the subducted slab, here the degree of eclogitization refers to the amount of eclogite in the lithospheric mantle. In our preferred model, the 7-km thick subducted oceanic crust becomes eclogite, while the lithospheric mantle constrains a different amount of eclogite."

*References:*

Faccincani, L., Faccini, B., Casetta, F., Mazzucchelli, M., Nestola, F. and Coltorti, M.: EoS of mantle minerals coupled with composition and thermal state of the lithosphere: Inferring the density structure of peridotitic systems, Lithos, 404–405(September), 106483, doi:10.1016/j.lithos.2021.106483, 2021.

Ganguly, J., Freed, A. M. and Saxena, S. K.: Density profiles of oceanic slabs and surrounding mantle: Integrated thermodynamic and thermal modeling, and implications for the fate of slabs at the 660km discontinuity, Phys. Earth Planet. Inter., 172(3–4), 257–267, doi:10.1016/j.pepi.2008.10.005, 2009.

Medaris, L. G., Brueckner, H. K., Cai, Y., Griffin, W. L. and Janák, M.: Eclogites in peridotite massifs in the Western Gneiss Region, Scandinavian Caledonides: Petrogenesis and comparison with those in the Variscan Moldanubian Zone, Lithos, 322, 325–346, doi:10.1016/j.lithos.2018.10.013, 2018.

Ohtani, E. and Maeda, M.: Density of basaltic melt at high pressure and stability of the melt at the base of the lower mantle, Earth Planet. Sci. Lett., 193(1–2), 69–75, doi:10.1016/S0012-821X(01)00505-2, 2001.

Robertson, E. C.: Thermal properties of rocks., 1988.

Skinner, B. J.: Section 6: Thermal expansion, in Handbook of Physical Constants, pp. 75–96, Geological Society of America., 1966.

---

## Author Comment (AC3)

General comments

The Tibetan Plateau is one of the most prominent collisional orogens around the world. Understanding the evolution and dynamics of Tibet provides crucial insights into the crust-mantle interactions and subducted processes. In the present study, the authors have conducted simultaneously high pressure-temperature experiments to constrain the density profiles of major minerals of eclogite (garnet, omphacite and epidote), and quantitatively unraveled the effect of density contrast on delamination of Tibet. They concluded that eclogite with a high garnet content and a high iron content and a high proportion of eclogite in the lithospheric mantle promote delamination. The experimental data are of high quality and convincing. Here I highly recommend it for Solid Earth.

Enclosed, please find our replies to the reviewer's comments in our revised manuscript entitled "Thermal equation of state of the main minerals of eclogite: Constraining the density evolution of eclogite during delamination process in Tibet".

We want to appreciate the reviewer for the thoughtful and thorough comments that have resulted in a substantial improvement in the revised version of our manuscript. Point-by-point responses to the reviewer's comments are shown in detail below. We have also indicated our changes are marked in blue in the revised manuscript. We believe that we have addressed all the reviewer's concerns adequately.

In the following, the reviewer's comments are shown in boldface and are followed by our replies in normal text. References cited are provided at the end of the response letter.

**Specific comments:**

**1. Comment: *Please specify the error in measuring temperature by using thermocouples in Section 3. Please add this information in supplementary tables as well. I wonder if the error in temperature is taken into account when fitting P-V- data to derive thermoelastic parameters.***

Thank you for your comment. For the electrical-resistance heating system, one of the major advantages is the stable and uniform T-distribution within the pressure chamber and the reliable control of temperatures by means of thermocouples (e.g., (Jenei et al., 2013; Miletich et al., 2009; Pasternak et al., 2008)). Moreover, the exceptional thermal conductivity of diamond as the anvil material has the major advantage of transferring heat to the sample inside the pressure chamber. Thus, the key points to obtain the stable and uniform T-distribution within the pressure chamber depend on the stabilization time at high temperature and the tightness of thermocouple glued to the diamond. Back to our high P-T experiments in this study, firstly, ensuring the thermocouple can correctly reflect temperature of sample chamber, we tightly glued the K-type thermocouples with the diamond. To minimize pressure instability and enhance temperature stability for each given heating run, the sample chamber was heated to a given temperature and then stabilize at least 5 minutes. Thus, we believe that the temperature error is very limited within our high P-T measurements in this study because the maximum temperature is just 700 K. This also can be confirmed by (Sinogeikin et al., 2006), who measured the temperature of sample chamber using two thermocouples attached to opposite sides of the diamond in a resistively heated DAC, which is similar with this study, and found the difference in temperature reading from these two thermocouples was within 1 K. To sum up, we did not consider the influence of the temperature error during the fitting process. We added the sentence at Line 170 in the revised manuscript:

"Before collecting data, the temperature in the sample chamber will be stabilized for 5 minutes and the temperature fluctuation is less than 1 K."

**2. Comment:** *In Section 4.1, the authors have first fitted the room-temperature data to obtain K0 and K0' at room temperature, and then used both room-temperature and high-temperature data to simultaneously derive K0, K0' and K0. Why did the authors not fix the room-temperature K0 and K0' values to get the thermal expansion parameter? What is the difference between two fitting methods?*

Thank you for your comment. The following Table 1 shows the thermal EoS parameters of eclogite rock-forming minerals obtained by fixing different parameters. The thermal expansions obtained by fixing different parameters are comparable within their uncertainties. This shows that the high temperature and high pressure experimental data obtained in this study are of high quality. Certainly, in our manuscript, the corresponding $K_{T0}$ and $K_T'_0$ are obtained by fixing $V_0$, which are obtained from the synchrotron single-crystal X-ray diffraction measurement at ambient conditions. Since the values from high-temperature and high-pressure data are similar to the results obtained by fitting the room-temperature and high-pressure data, we do not fix the room-temperature $K_{T0}$ and $K_T'_0$ values to get the thermal expansion parameter.

**Table 1.** Thermal EoS parameters derived from the fitting of *P-V-T* data to the BM3-HP=Third-order Birch-Murnaghan compressional EoS in combination with the Holland Powell thermal-pressure EoS.

| Composition | $V_0$ (Å$^3$) | $K_{T0}$ (GPa) | $K_T'_0$ | $\alpha_0$ (10$^{-5}$ K$^{-1}$) | $\theta_E$ | |
|---|---|---|---|---|---|---|
| Garnet (Prp$_{21}$Alm$_{47}$Grs$_{31}$Sps$_1$) | 1566.05(25) | 170(1) | 3.74(22) | | | |
| | 1566.05(25) | 170(1) | 3.82(14) | 2.71(5) | 450[a] | Fixed $V_0$ |
| | 1566.05(25) | 170(1) | 3.74(22) | 2.74(3) | 450[a] | Fixed $V_0$, $K_{T0}$, and $K_T'_0$ |
| Omphacite (Quad$_{48}$Jd$_{45}$Ae$_7$) | 423.48(24) | 121(2) | 3.90(35) | | | |
| | 423.48(24) | 121(3) | 3.97(34) | 3.73(20) | 343[a] | Fixed $V_0$ |
| | 423.48(24) | 121(2) | 3.90(35) | 4.12(11) | 343[a] | Fixed $V_0$, $K_{T0}$, and $K_T'_0$ |
| Epidote (Ca$_{2.02}$Fe$_{0.75}$Al$_{2.32}$Si$_{0.16}$[SiO$_4$][Si$_2$O$_7$] O(OH)) | 461.57(23) | 122(1) | 2.51(16) | | | |
| | 461.57(23) | 124(2) | 2.04(15) | 3.04(13) | 626[b] | Fixed $V_0$ |
| | 461.57(23) | 122(1) | 2.51(16) | 2.67(11) | 626[b] | Fixed $V_0$, $K_{T0}$, and $K_T'_0$ |

**3. Comment:** *In Section 4.2, the authors have discussed the compositional effect on the elasticity of major minerals of eclogite. One of the main conclusions is that the incorporation of iron would reduce the bulk modulus of omphacite. Do the ferrous iron and the ferric iron impose a comparable effect on the bulk modulus of omphacite?*

Thanks for the constructive advice. We collected the isothermal bulk modulus and its pressure derivative of clinopyroxenes as shown in Table 2. The bulk modulus of Fe-free jadeite is between 125-137 GPa, which is larger than 115-123.6 GPa of other clinopyroxene containing Fe. The addition of iron does reduce the bulk modulus. We collected the bulk modulus of omphacite containing ferrous iron and ferric iron as shown in the figure 1 below. There is no obvious relationship between the bulk modulus of omphacite and the content of ferric iron. The bulk modulus

of Fe-bearing clinopyroxene is between 115-123.6 GPa, but the ferric iron does not show a comparable effect on the bulk modulus. At least from the data we have collected, the effect of ferric iron is limited, and no significant effect on bulk moduli of omphacite can be seen.

[Figure]

**Figure 1**. Isothermal bulk modulus and $Fe^{3+}$ content of omphacites and other clinopyroxenes including hedenbergite and aegirine.

**Table 2.** Isothermal bulk modulus and its pressure derivative of omphacites and other clinopyroxenes including hedenbergite and aegirine.

| Composition | $Fe^{3+}\%$ | $V_0$ (Å³) | $K_{T0}$ (GPa) | $K_{T'0}$ | References |
|---|---|---|---|---|---|
| $Jd_{100}$ | Fe-free | 403.32(8) | 125(4) | $5^{fixed}$ | (Zhao et al., 1997) |
| $Jd_{100}$ | Fe-free | 402.26(2) | 134.0(7) | 4.4(1) | (Nestola et al., 2006) |
| $Jd_{100}$ | Fe-free | 402.03(2) | 137(1) | 3.4(4) | (McCarthy et al., 2008) |
| $Jd_{100}$ | Fe-free | 402.5(4) | 136(3) | 3.3(2) | (Posner et al., 2014) |
| $Hd_{100}$ | 0 | 449.90(7) | 117(1) | 4.3(4) | (Zhang et al., 1997) |
| $Quad_{57}Jd_{42}Ae_1$ | 1.7 | 422.3(1) | 123.6(5) | $4_{fixed}$ | (Xu et al., 2019) |
| $Quad_{52}Jd_{44}Ae_4$ | 7.1 | 423.9(3) | 116(2) | 4.3(2) | (Zhang et al., 2016) |
| $Quad_{49}Jd_{45}Ae_6$ | 10.9 | 422.2(1) | 117(3) | 6.0(6) | (Pavese et al., 2001) |
| $Quad_{48}Jd_{45}Ae_7$ | 12.7 | 423.48(24) | 121(3) | 3.97(34) | This study |
| $Quad_{53}Jd_{27}Ae_{20}$ | 27.4 | 426.0(2) | 115(2) | 4.9(4) | (Xu et al., 2019) |
| $Ae_{100}$ | 100 | 431.5(1) | 118(3) | 4.3(3) | (Xu et al., 2017) |
| $Ae_{100}$ | 100 | 429.26(2) | 116.1(5) | 4.4(1) | (Nestola et al., 2006) |
| $Ae_{100}$ | 100 | 429.40(9) | 117(1) | 3.2(2) | (McCarthy et al., 2008) |

**4. Comment:** *In Section 5, the authors have modeled the density of eclogite with varying amounts of garnet to explore the effect of mineral compositions on the density of eclogite. What are the partitioning behaviors of ferrous iron and ferric iron in coexisting garnet, omphacite and epidote? Do they alter as a function of pressure (or depth)? I notice that the iron component has distinct effects on the thermoelasticity of these minerals. Does the distribution of iron affect the delamination?*

Thanks for the constructive advice. The partitioning behaviors of ferrous iron and ferric iron in

eclogite rock forming minerals have not been systematically considered before. According to the pressure and temperature conditions of Paleozoic (210-280 Ma) eclogite (Cheng et al., 2012; Tang et al., 2013, 2020; Yang et al., 2009; Zhu et al., 2015) calculated by geothermometer Grt-Omp, the obtained pressures are usually 2.4-3.6 GPa, and the corresponding depth is about 70-120 km. Through specific EMPA data and corresponding P-T conditions, we plotted the relationship of the ferric iron and ferrous iron in garnet and omphacite with depth in eclogites. Since the composition of epidote is limited, it is difficult to consider Fe in epidote here. The ratio of ferric iron and ferrous iron in garnet and omphacite does not show a significant relationship with depth (pressure). For garnet and omphacite in the same location with similar temperature and pressure conditions, the content of ferric iron or ferrous iron in them is different. Moreover, the partitions of ferric and ferrous irons in garnet and omphacite also do not appear to be significantly related to depth (pressure), at least from the data we collected. As far as the composition of Tibetan eclogite is concerned, the iron content does not change regularly with depth. To solve this problem, more natural eclogite rock-forming minerals may be needed to conduct a more systematic calculation of the temperature and pressure conditions, and specific composition.

[Figure]

**Figure 2**. The distribution of ferric iron/ ferrous iron ($Fe^{3+}$/ $Fe^{2+}$) in garnet (a) and omphacite (b) with depth. Proportions of ferric (c) and ferrous (d) irons in garnet and omphacite, respectively, as a function of depth.

**5. Comment: *In Section 5, "10% increase in garnet" should be "10 vol% increase in garnet".***
Thank you for your comment. We revised "10% increase in garnet" to "10 vol% increase in garnet" in Line 385 and Line 396.

**6. Comment:** *The references of this manuscript is over-cited. The references that are not closely related to the present study could be removed.*

Thank you for your comment. We checked references in our manuscript and removed some less closely related articles.

***References:***

Cheng, H., Zhang, C., Vervoort, J. D., Lu, H., Wang, C. and Cao, D.: Zircon U–Pb and garnet Lu–Hf geochronology of eclogites from the Lhasa Block, Tibet, Lithos, 155, 341–359, doi:10.1016/j.lithos.2012.09.011, 2012.

Jenei, Z., Cynn, H., Visbeck, K. and Evans, W. J.: High-temperature experiments using a resistively heated high-pressure membrane diamond anvil cell, Rev. Sci. Instrum., 84(9), 095114, doi:10.1063/1.4821622, 2013.

McCarthy, A. C., Downs, R. T., Thompson, R. M. and Redhammer, G. J.: In situ high-pressure single-crystal X-ray study of aegirine, $NaFe3+Si2O6$, and the role of M1 size in clinopyroxene compressibility, Am. Mineral., 93(11–12), 1829–1837, doi:10.2138/am.2008.2725, 2008.

Miletich, R., Cinato, D. and Johänntgen, S.: An internally heated composite gasket for diamond-anvil cells using the pressure-chamber wall as the heating element, High Press. Res., 29(2), 290–305, doi:10.1080/08957950902747403, 2009.

Nestola, F., Boffa Ballaran, T., Liebske, C., Bruno, M. and Tribaudino, M.: High-pressure behaviour along the jadeite $NaAlSi2O6$–aegirine $NaFeSi2O6$ solid solution up to 10 GPa, Phys. Chem. Miner., 33(6), 417–425, doi:10.1007/s00269-006-0089-7, 2006.

Pasternak, S., Aquilanti, G., Pascarelli, S., Poloni, R., Canny, B., Coulet, M.-V. and Zhang, L.: A diamond anvil cell with resistive heating for high pressure and high temperature x-ray diffraction and absorption studies, Rev. Sci. Instrum., 79(8), 085103, doi:10.1063/1.2968199, 2008.

Pavese, A., Diella, V., Pischedda, V., Merli, M., Bocchio, R. and Mezouar, M.: Pressure-volume-temperature equation of state of andradite and grossular, by high-pressure and -temperature powder diffraction, Phys. Chem. Miner., 28(4), 242–248, doi:10.1007/s002690000144, 2001.

Posner, E. S., Dera, P., Downs, R. T., Lazarz, J. D. and Irmen, P.: High-pressure single-crystal X-ray diffraction study of jadeite and kosmochlor, Phys. Chem. Miner., 41(9), 695–707, doi:10.1007/s00269-014-0684-y, 2014.

Sinogeikin, S., Bass, J., Prakapenka, V., Lakshtanov, D., Shen, G., Sanchez-Valle, C. and Rivers, M.: Brillouin spectrometer interfaced with synchrotron radiation for simultaneous x-ray density and acoustic velocity measurements, Rev. Sci. Instrum., 77(10), 103905, doi:10.1063/1.2360884, 2006.

Tang, Y.-J. J., Zhang, H.-F. F., Ying, J.-F. F., Su, B.-X. X., Chu, Z.-Y. Y., Xiao, Y. and Zhao, X.-M. M.: Highly heterogeneous lithospheric mantle beneath the Central Zone of the North China Craton evolved from Archean mantle through diverse melt refertilization, Gondwana Res., 23(1), 130–140, doi:10.1016/j.gr.2012.01.006, 2013.

Tang, Y., Qin, Y., Gong, X., Duan, Y., Chen, G., Yao, H., Liao, J., Liao, S., Wang, D. and Wang, B.: Discovery of eclogites in Jinsha River suture zone, Gonjo County, eastern Tibet and its restriction on Paleo-Tethyan evolution, China Geol., 3(1), 83–103, doi:10.31035/cg2020003, 2020.

Xu, J., Zhang, D., Fan, D., Downs, R. T., Hu, Y. and Dera, P. K.: Isosymmetric pressure-induced bonding increase changes compression behavior of clinopyroxenes across jadeite-aegirine solid solution in subduction zones, J. Geophys. Res. Solid Earth, 122(1), 142–157,

doi:10.1002/2016JB013502, 2017.

Xu, J., Zhang, D., Fan, D., Dera, P. K., Shi, F. and Zhou, W.: Thermoelastic Properties of Eclogitic Garnets and Omphacites: Implications for Deep Subduction of Oceanic Crust and Density Anomalies in the Upper Mantle, Geophys. Res. Lett., 46(1), 179–188, doi:10.1029/2018GL081170, 2019.

Yang, J., Xu, Z., Li, Z., Xu, X., Li, T., Ren, Y., Li, H., Chen, S. and Robinson, P. T.: Discovery of an eclogite belt in the Lhasa block, Tibet: A new border for Paleo-Tethys?, J. Asian Earth Sci., 34(1), 76–89, doi:10.1016/j.jseaes.2008.04.001, 2009.

Zhang, D., Hu, Y. and Dera, P. K.: Compressional behavior of omphacite to 47 GPa, Phys. Chem. Miner., 43(10), 707–715, doi:10.1007/s00269-016-0827-4, 2016.

Zhang, L., Ahsbahs, H., Hafner, S. S. and Kutoglu, A.: Single-crystal compression and crystal structure of clinopyroxene up to 10 GPa, Am. Mineral., 82(3–4), 245–258, doi:10.2138/am-1997-3-402, 1997.

Zhao, Y., Von Dreele, R. B., Shankland, T. J., Weidner, D. J., Zhang, J., Wang, Y. and Gasparik, T.: Thermoelastic equation of state of jadeite NaAlSi 2 O 6 : An energy-dispersive Reitveld Refinement Study of low symmetry and multiple phases diffraction, Geophys. Res. Lett., 24(1), 5–8, doi:10.1029/96GL03769, 1997.

Zhu, D.-C., Wang, Q., Zhao, Z.-D., Chung, S.-L., Cawood, P. A., Niu, Y., Liu, S.-A., Wu, F.-Y. and Mo, X.-X.: Magmatic record of India-Asia collision, Sci. Rep., 5(1), 14289, doi:10.1038/srep14289, 2015.